# The Urinary Microbiome in Health and Disease: Relevance for Bladder Cancer

**DOI:** 10.3390/ijms25031732

**Published:** 2024-01-31

**Authors:** Natasa Kustrimovic, Giorgia Bilato, Lorenzo Mortara, Denisa Baci

**Affiliations:** 1Center for Translational Research on Autoimmune and Allergic Disease—CAAD, Università del Piemonte Orientale, 28100 Novara, Italy; natasa.kustrimovic@gmail.com; 2Immunology and General Pathology Laboratory, Department of Biotechnology and Life Sciences, University of Insubria, 21100 Varese, Italy; gbilato@uninsubria.it; 3Molecular Cardiology Laboratory, IRCCS—Policlinico San Donato, 20097 Milan, Italy

**Keywords:** bladder cancer, genitourinary microbiome, inflammation

## Abstract

Bladder cancer (BC) constitutes one of the most diagnosed types of cancer worldwide. Advancements in and new methodologies for DNA sequencing, leading to high-throughput microbiota testing, have pinpointed discrepancies in urinary microbial fingerprints between healthy individuals and patients with BC. Although several studies suggest an involvement of microbiota dysbiosis in the pathogenesis, progression, and therapeutic response to bladder cancer, an established direct causal relationship remains to be elucidated due to the lack of standardized methodologies associated with such studies. This review compiles an overview of the microbiota of the human urinary tract in healthy and diseased individuals and discusses the evidence to date on microbiome involvement and potential mechanisms by which the microbiota may contribute to the development of BC. We also explore the potential profiling of urinary microbiota as a biomarker for risk stratification, as well as the prediction of the response to intravesical therapies and immunotherapy in BC patients. Further investigation into the urinary microbiome of BC patients is imperative to unravel the complexities of the role played by host–microbe interactions in shaping wellness or disease and yield valuable insights into and strategies for the prevention and personalized treatment of BC.

## 1. Introduction

Bladder cancer (BC) ranks as the 11th most frequently diagnosed typology of cancer globally, characterized by high incidence and mortality rates [1]. It is one of the most common tumors of the urological tract, second only to prostate cancer in Europe and the US [2]. Environmental factors, including smoking and occupational exposure, alongside genetic factors, contribute to BC development [3,4].

BC is typically classified into muscle-invasive bladder cancer (MIBC), non-muscle-invasive bladder cancer (NMIBC), and metastatic tumor from distant sites. Around 70% of BC occurs in the cells of the bladder’s innermost lining, i.e., the urothelium. NMIBC are less lethal and are easier to treat, usually by transurethral resection, even though the recurrence rates are quite high (40%–80%) [5]. In some instances, chemotherapy or radiotherapy can be included, as well as immunotherapy by Bacillus Clemette Guerin (BCG) vaccine, raising the five-year survival rate to around 90% [6].

The tumors that have grown into the muscle wall of the bladder (MIBC) constitute about 30% of the cases and are highly prone to metastasis [7]. MIBC has few early symptoms, progresses rapidly, and has a poor prognosis, with a 5-year survival rate lower than 50% in the stage 3 [8]. Despite availability of radical cystectomy and pelvic lymph node dissection, approximately half of the patients with deep MIBC involving the muscularis propria develop a metastatic disease within 2 years of diagnosis [9]. Therefore, systemic therapy plays a key role in conjunction with local therapy to reduce rates of recurrence. Neoadjuvant chemotherapy (NCT) has been attracting substantial attention and has been extensively evaluated. Several randomized clinical trials have shown that platinum-based combination neoadjuvant chemotherapy applied before the surgery can significantly improve survival outcomes [10]. The adoption of immune checkpoint inhibitors (ICIs) represents a crucial and evolving approach to overcome challenges associated with the provision of impactful neoadjuvant or adjuvant systemic therapy to a significant portion of the patients with BC. PD-1/PD-L1 inhibitors have already received approval for specific applications in advanced BC [11].

In recent years, there has been considerable evolution in understanding the factors that cause cancer and their impact on its development, leading to significant improvements in incidence and mortality worldwide [12]. Despite these advancements in diagnostic methods and therapeutic procedures, an unmet clinical need persists for early diagnostic tools that could guide effective treatment strategies. BC, like the majority of other cancers, remains incurable in advanced stages. Improving our understanding of BC development and progression (including premalignant lesions) could address some of the unmet needs faced by affected patients. Exploring potential clues in less evident sites is crucial for advancing this understanding. Growing evidence over recent years suggests that the therapeutic response to systematic treatment potentially lies in the link with the genitourinary microbiota. Over recent years, the direct or indirect relationship between cancer and the specific microflora of different cancer types, including BC, has attracted intense research attention [5].

The human microbiome is composed of bacteria, bacteriophages, viruses, fungi, and protozoa that are located in the epithelial surfaces of several body areas: the genitourinary tract, the skin, the oral cavity, and the gastrointestinal tract. Each of these areas exhibits considerable interindividual differences in microbial composition, rendering them distinct entities [13].

Nowadays, it is a confirmed fact that the microbiome affects several important physiological functions, such as inflammation, metabolism, hematopoiesis, as well as cognitive abilities [14]. Moreover, the microbiome can impact cancer development and response to therapies through mechanisms such as direct damage to host DNA by released bacterial toxins, inflammation-induced cancer transformation, alteration of the cellular microenvironment mediated by the microbiome, and the production of microbial metabolites with potential carcinogenic effects [15,16].

Bacteria, fungi, or viruses in the genitourinary tract can be a potential cause of initiation and/or progression of urological tumors [17]. Conversely, the microbiota can play a significant role in the treatment of these cancers, as exemplified by the BCG vaccine widely used to prevent the recurrence of high-risk NMIBC [18].

Nonetheless, we have so far only scratched the surface regarding the potential role of microbiota in the tumorigenesis, progression, immunomodulation, and drug efficacy of BC. In this review, we aim to provide a summary of current findings on the potential influence of the microbiota in bladder cancer development and progression, how specific bacteria are implicated in this type of cancer, and, finally, how the microbiome can impact treatments.

## 2. The Genitourinary Microbiome 

It is noteworthy that, when the Human Microbiome Project (HMP) was initiated in 2008 with the goal of comprehensively characterizing the human microbiome, the investigation of the bladder microbiome was excluded. This exclusion was primarily due to ethical considerations, as obtaining satisfactory samples with minimal contamination from the urethra would require invasive techniques (bladder biopsy or suprapubic aspirate, SPA) applied to healthy individuals [19]. The most appropriate sample for identifying the BC microbiome is thought to be urine, since it is stored in the bladder. However, a significant issue arises regarding urine sample collection, as different methods (e.g., midstream urine collection, MSU, catheterization with an intermittent or permanent catheter, TUC, or SPA) can lead to variations in the microbiota [20]. Differences among these methods lie in the level of invasion associated with the methodology, with MSU collection being non-invasive and SPA being highly invasive. Additionally, the degree of bacterial contamination from other areas, such as the urethra, skin, or genital apparatus, varies (MSU having the highest risk of contamination and SPA the lowest) [21]. The second criteria for the exclusion of the bladder microbiome from the HMP investigation was the long-standing belief that the urine of healthy individuals was sterile. This belief was based on findings from traditional microbiological experiments, which were limited in their ability to detect microorganisms. These methods primarily identified aerobic and fast-growing bacteria, such as *Escherichia coli*, while anaerobic microorganisms with slow growth or bacteria with complex nutrient requirements remained undetected [21].

With the progressive development of diverse diagnostic technologies, in particular next-generation sequencing technologies (16S rRNA sequencing and 18S rRNA sequencing, among others), that assumption has been challenged, and we now know that urine has a unique microbiome composition that is influenced by a number of elements, such as gender, age, sexual behavior, and concomitant diseases [19,22,23].

Currently, there is a lack of standardized procedures for urine collection, preservation, and storage in microbiome research. Studies have explored various storage conditions, including temperature, storage time, and the addition of preservatives. Results indicate that lower temperatures, shorter storage times, and the use of preservatives contribute most effectively to the reproducibility of the urine microbiome [24].

### 2.1. The Healthy Urinary Microbiome 

The term “urobiome,” frequently used to describe the microbial community in bladder-obtained urine, is used interchangeably with “urinary microbiota” [25]. The urinary microbiome encompasses the genes, genomes, and metabolites of the microbiota and host environment. Variations in the urinary microbiome can be characterized in terms of α diversity (diversity of populations in a sample) and β diversity (population differences between samples) [26].

The microbiota found in urine is generally less abundant and less diverse compared to the microbiota in other body sites, such as the gut. In the female urobiome, there are approximately 10^4^–10^5^ colony-forming units (CFU)/mL, in contrast to 10^12^ CFU/g in the feces [27]. To date, more than 100 species from more than 50 genera have been identified in the human genitourinary tract [28,29,30].

Interestingly, the urinary microbiome in males and females is similar, with very few differences. In both genders, the majority of microbiome species (97%) belongs to five major phyla, with the phylum Firmicutes (65% in males vs. 73% in females) being the most prevalent [21]. The remaining phyla are Actinobacteria (15% males vs. 19% females), Bacteroidetes (10% males vs. 3% females), Proteobacteria (8% males vs. 3% females), and Fusobacteria [31,32]. The genera found in both females and males include *Lactobacillus*, *Corynebacterium*, *Prevotella*, *Staphylococcus*, *Streptococcus*, *Escherichia*, *Enterococcus*, and *Citrobacter* [32,33]. The genera *Lactobacillus* and *Gardnerella* are predominant in the female microbiota, whereas the male microbiota presents a higher percentage of *Corynebacterium*, *Staphylococcus*, and *Streptococcus* [34,35], while *Pseudomonas* has been detected only in men [32]. However, there is significant interindividual variability, leading to the presence of still undefined members of a core genitourinary tract microbiome.

#### 2.1.1. The Urinary Microbiome in Healthy Females 

Urinary bacterial populations have been grouped into diverse urotypes in which a particular bacterial genus predominates [27,36]. The most abundant genus found in the urinary microbiome of healthy women is *Lactobacillus* [27,29,30,31,32,36,37,38,39,40,41,42,43,44,45]. *Lactobacillus* is an acid-producing, facultative anaerobic bacteria known to play protective roles in the vaginal tract by decreasing the pH and producing various bacteriostatic/bactericidal compounds [46,47]. The second most frequently isolated bacterial strain in the urine of healthy women is *Gardnerella*, found in abundance by numerous studies, with *Gardnerella vaginalis* representing the most abundant species [27,29,30,31,32,36,38,39,41,42,43,44,48].

Beside *Lactobacillus* and *Gardnerella*, other urotypes identified in the urinary microbiome of healthy females are: *Prevotella*, *Corynebacterium*, *Sneathia*, *Streptococcus*, and *Escherichia* urotypes. Other genera also identified include *Aerococcus*, *Alloscardovia*, *Anaerococcus*, *Bifidobacterium*, *Corynebacterium*, *Enterococcus*, *Finegoldia*, *Klebsiella*, and *Staphylococcus* (Table 1).

As of today, age has been infrequently investigated as a factor in the change of urinary microbiome. One study found that, at the bacterial phylum level, samples from elderly individuals were dominated by Proteobacteria, Firmicutes, Bacteroidetes, Actinobacteria, and Thermi, while samples from non-elderly individuals exhibited a similar dominance with slight variations. At the genus level, both cohorts showed predominance of *Prevotella*, followed by *Bacteroides*, *Lactobacillus*, *Pseudomonas*, and *Acinetobacter* [49]. Interestingly, the elderly cohort displayed a lower incidence of *Lactobacillus* [49,50]. Recently, the *Escherichia* urotype has been linked to healthy elderly women, whereas the *Gardnerella* urotype has been associated with young women [44]. 

An intriguing question surrounds the origin of the urinary microbiota, with the vagina and gut being proposed as potential sources. Studies have reported a significant similarity between the urinary and vaginal microbiomes of healthy females, including both uropathogens (*E. coli* and *Streptococcus anginosus*) and commensal bacteria (*L. iners* and *L. crispatus*) [51]. Nevertheless, differences such as the absence of the genera *Tepidomonas* and *Flavobacterium* in the vaginal microbiota have also been noted [43]. Significant overlap (64%) between gut microbiota and urinary microbiota has been reported [52,53].

The origin of the urinary microbial community remains unclear, and both hypotheses, advancing both the vagina and the gut as potential sources, seem plausible at this moment. However, more data are needed to attain a conclusive understanding in this regard.

#### 2.1.2. The Urinary Microbiome in Healthy Males 

Studies examining the urinary microbiome of healthy males are far less numerous than those focusing on females. Early studies of the healthy male urinary microbiome identified genera such as *Lactobacillus*, *Sneathia*, *Veillonella*, *Corynebacterium*, *Prevotella*, *Streptococcus*, and *Ureaplasma* [54,55]. A study of men with and without sexually transmitted infections (STIs) found that bacteria associated with STI, such as *Sneathia*, *Mycoplasma*, and *Ureaplasma*, are present in urine samples of STI-positive patients [37] or sexually active males [55]. It is important to mention that those first studies were exclusively conducted on FC urine samples or urethral swabs, thus introducing risks of potential contamination. A study from 2020 clearly emphasized the importance of using catheterized urine samples in male urobiome studies [56].

Predominant urotypes found in healthy males are similar to those found in females: *Prevotella*, *Shigella*, *Enterococcus*, *Streptococcus*, and *Citrobacter* [36], with predominance of *Corynebacterium* [38] and *Streptococcus* [32]. Other described members of a healthy male urobiome are *Lactobacillus* and *Pseudomonas* [57], *Staphylococcus haemolyticus* [58], and coagulase-negative *Staphylococci* and *Eubacterium* [40] (Table 2).

### 2.2. The Urinary Microbiome of Diseased Individuals 

Dysbiosis of urinary microbiota is closely linked to the development or progression of several urinary diseases, such as urinary tract infections (UTIs), interstitial cystitis (IC), urinary incontinence (UI), and bladder pain syndrome (BPS).

Bacteria reside around the urethra and can colonize the bladder, although they are expelled during micturition. UTIs are more common in women than in men due to their anatomical characteristics. The shorter urethra, the proximity of the urethral opening to the bladder, and the close proximity to the rectum makes it easier for bacteria to reach the bladder in women. Furthermore, urogenital manipulations associated with daily living, sexual intercourse, or medical interventions facilitate the movement of bacteria toward the urethra [59]. Bacterial colonization typically begins in the urethral cavity, causing the infection of the lower urinary tract, but can progress to the bladder, thus causing cystitis [60]. In rare cases, the infection can progress to the kidneys, causing pyelonephritis, and even spread through the bloodstream, leading to a systemic infection (urosepsis) [60].

Bacteria that have been commonly associated with UTIs are *Escherichia coli* (found in 80% of the cases), *Enterococcus*, and *Staphylococcus* [61]. *Escherichia coli* is part of the commensal urinary microbiome and various factors may contribute to its involvement in UTIs [62]. It has been shown that *E. coli* can exhibit higher pathogenicity in polymicrobial infections, especially with *Enterococcus*, though the underlying mechanisms remain to be fully elucidated [63]. Additionally, certain bacteria from healthy urinary commensal microbiota, such as *Corynebacterium glucuronolyticum*, *Streptococcus gallolyticus*, and *Aerococcus sanguinicola*, are known to cause infections [21]. These findings suggest that some UTIs may result from an imbalance in the urinary microbiota repertoire rather than from invasion by an external pathogenic organism.

Uncontrolled loss of urine, or urinary incontinence (UI), can be categorized into stress urinary incontinence (SUI), urgency urinary incontinence (UUI), and mixed urinary incontinence (MUI) [64]. Patients with UUI present a lower abundance of *Lactobacillus* and a higher abundance of *Gardnerella*, along with other genera such as *Actinobaculum*, *Actinomyces*, *Aerococcus*, *Arthrobacter*, *Corynebacterium*, *Oligella*, *Staphylococcus*, and *Streptococcus* [27]. In addition, increased abundance of *Lactobacilus gasseri* has been reported [27]. A study from 2016 analyzing the urinary microbiome of UUI patients identified certain enriched genera, such as *Methylobacterium*, *Brevundimonas*, *Chitinophaga*, and *Sphingomonadales*, while some genera were not abundant: *Prevotella*, *Comamonadaceae*, *Nocardioides*, and *Mycobacterium* [42]. Conversely, some of the most abundant genera identified in the study performed by Pearce et al. [27] were not found in this study, which implies the necessity for ulterior research regarding this topic. In patients with MUI, a decreased relative abundance of *Lactobacillus* and an increase in *Gardenerella* and *Prevotella* were found [50], while in patients with SUI no differences in urinary microbiome were found [65].

In patients with overactive bladder syndrome (OAB), a lower abundance of *Lactobacillus* was detected, alongside a higher abundance of *Proteus* [66,67]. In addition, abundance of *Atopobium vaginae* and *Finegoldia magna* was reported as well [68].

The analysis of mid-stream urine from female patients with interstitial cystitis/bladder pain syndrome has shown differences in the abundance of *Lactobacillus gasseri* and a reduction in *Corynebacterium*, with the exclusive presence of *Proteus mirabilis*, *Pseudomonas aeruginosa*, *Francisella tularensis*, *Mycoplasma hyorhinis*, *Helicobacter hepaticus*, and *Clostridium perfringens* [69,70]. Nevertheless, recent studies using 16S rRNA gene sequencing found no differences in the urinary microbiota of individuals affected by this syndrome [71,72].

Patients affected by chronic prostatitis (CP)/chronic pelvic syndrome (CPS) have shown increased abundance of *Clostridia*, *Bacteroides*, or *Porphyromonas* and a decreased presence of *Bacilli* [73].

A study on the microbiome of patients with acute uncomplicated cystitis (AUC) recorded the presence of *Pseudomonas*, *Acinetobacter*, and *Enterobacteriaceae*, while the recurrent cystitis (RC) group was characterized by the presence of *Sphingomonas*, *Staphylococcus*, *Streptococcus*, and *Rothia* spp. [74].

Regarding the microbiome associated with described disease states of the urinary tract (UTI, UI, OAB, IC, BPS, CP, CPS), no single microorganism has been clearly identified as a specific causative agent for the particular pathology; however, there are strong signals that an altered microbiome or global shifts in bacterial communities (urotypes) may be implicated in disease initiation and progression, although definitive conclusions cannot be drawn just yet.

## 3. The Bladder Cancer Microbiome

In most urinary microbiome studies, urine collected using different methods have provided valuable insights into the microbial community present in the urinary bladder. However, important questions have arisen: does the urinary microbiome adequately represent the bladder microbiome? Could there be bladder mucosa-associated bacteria not detected in urine samples? Would it be more appropriate to examine bladder tissue to reduce potential contamination risks? Unfortunately, studies on bladder tissue microbiome are scarce due to the extremely invasive nature of bladder tissue biopsy. The overall picture is further complicated by differences between fresh-frozen (FF) tissue and formalin-fixed paraffin-embedded (FFPE) tissue, a topic that requires further investigation (reviewed in Zhang et al. [5]).

In 2020, Pederzoli et al. demonstrated that the microbiome identified in the urine of BC patients exhibited over 80% similarity with the microbiome found within BC tissues, concluding that urinary microbiome may correctly reflect the microbiome in the BC environment [75]. Nevertheless, another group of authors have found abundance of *Akkermansia*, *Bacteroides*, *Clostridium sensu stricto*, *Enterobacter*, and *Klebsiella* in tissue samples compared to the urine samples of BC patients [76]. This leaves the question of the adequacy of investigated samples open for further discussion.

The bladder microbiome may promote or inhibit BC pathogenesis and progression (Figure 1). The potential relationships between urinary tract microorganisms and BC have been identified in several instances. A variety of bacteria produce alkaline proteases that can act intracellularly and/or extracellularly, leading to changes in the extracellular matrix and influencing its renewal, thus playing an important role in host tissue degradation and immune system evasion and/or destruction of host physical barriers, in turn facilitating pathogen spread through the body and contributing to the onset of infections [77,78]. It has been speculated that in the bladder this can result in an altered and potentially cancer-promoting extracellular milieu [77,79].

A summary of the results from the available studies linking the prevalence of certain bacteria in BC is given in Table 3. Even though these studies have identified several organisms exhibiting a significant association with BC, the overall results are not coherent among them, thus making it impossible for us to draw definitive conclusions. No difference in the α diversity of urinary microbiota in BC patients has been found [80], while the β diversity has been found to be elevated in BC samples [81]. A reduction in bacterial richness and diversity in BC patients has been confirmed by several other studies [82,83].

One of the first studies investigating the urinary microbiome and BC correlation from bladder mucosa tissue found that the Shannon diversity index was significantly lower in the BC tissue, suggesting a significantly lower degree of bacterial diversity in cancer tissues [82]. It is notable that a loss of microbiota diversity has also been described in the case of colorectal cancer [84]. Liu et al. have detected lower relative abundances of *Lactobacillus*, *Prevotella_9*, as well as Ruminococcaceae in BC tissues, whereas *Cupriavidus* spp., Brucellaceae, *Acinetobacter*, *Anoxybacillus*, *Escherichia-Shigella*, *Geobacillus*, *Pelomonas*, *Ralstonia*, and *Sphingomonas* have been found to occur in higher numbers. These findings have led the authors to speculate that these genera may be utilized, potentially, as biomarkers for bladder cancer [82].

In urine samples collected from male patients with BC, Firmicutes was the most abundant phylum, followed by Actinobacteria, Bacteroidetes, and Proteobacteria [80]. Significantly higher relative levels of *Acinetobacter*, *Anaerococcus*, *Rubrobacter*, *Sphingobacterium*, *Atoposites*, and *Geobacillus* in the urine of BC patients have been reported as well [85]. Furthermore, a high abundance of *Bacteroides* and *Faecalbacterium* species has been found in cancer samples [83].

Interestingly enough, *Ruminococcus* and *Bifidobacterium*, genera known to be anti-inflammatory and important in mucosal homeostasis, have been found to occur in lower amounts in BC patients [86,87,88]. A decrease in beneficial bacteria such as these may lead to the creation of an environment that quickens inflammation and oxidative stress, inflicting damage on bladder tissue.

Abundance of Firmicutes in urine and tissue samples collected from BC patients has been reported recently, with the addition of Cyanobacteria as a significant phylum [76]. The latter is particularly important, since it has been confirmed that toxic products from Cyanobacteria, i.e., microcystins, can induce hepatocellular cancer and promote migration and invasion of colorectal cancer [89]. In addition, *Streptococcus*, *Corynebacterium*, and *Fusobacterium* genera have also been found to be abundant in the urine of cancer patients [76]. The same group of researchers further analyzed tissue samples of BC patients and found high interindividual variability among the microbiota characterizing the tissue samples. However, five genera, namely *Akkermansia*, *Bacteroides*, *Clostridium*, *Enterobacter*, and *Klebsiella* were over-represented in the tissue samples compared to the urine samples [76]. This difference indicates that these genera are directly associated with the tissue, but further studies with larger sample sizes of cancerous bladder mucosa and adjacent normal healthy tissues are needed.

Importantly enough, all the bacterial phyla that have been found to occur abundantly in the bladder tissue of BC patients are known participants in the development of several forms of cancer. *Bacteroides* and *Akkermansia* are strongly positively correlated with an increased tumor burden in colorectal cancer [90] since they are known mucin degraders, undermining the integrity of the mucosal barrier and leading to increased inflammation [91,92]. *Enterobacter* spp. may constitute a clinically important factor for colon cancer initiation and progression due to apoptosis inhibition [93]. *Klebsiella pneumoniae* produces the colibactin toxin known to cause DNA double-strand breaks, inducing genomic instability as well as cell cycle arrest [94]. In addition, the colibactin toxin leads to chronic inflammation and stimulation of epithelial cell proliferation in the microenvironment of the colon [95]. On the other hand, *Clostridium* association with malignant transformation is species-dependent [96].

In several of the studies thus far available, abundance of *Acinetobacter* and *Proteobacteria* has been found in BC specimens. *Acinetobacter* has been found to contribute to the development of multidrug resistance and cause a variety of diseases, including pneumonia and bloodstream infections [97], while the dysbiosis of the genus *Proteobacteria* has been found to correlate with Crohn’s disease and colitis-associated colorectal cancer [98,99]. Therefore, an increase in *Acinetobacter* and *Proteobacteria* could be interpreted as a dysbiosis marker for the diagnosis of BC.

Interestingly enough, in several instances, the role of gender differences has also been evaluated. The incidence of BC has been found to be higher in men, while women have poorer outcomes and higher death rates [100,101], thus emphasizing the potential influence of sex hormones on BC occurrence and progression and drawing a parallel between sex-associated differences in the urinary microbiome [101,102]. As already discussed above, a healthy individual’s urinary microbiome shows variations between males and females. When it comes to BC patients, results have shown that tissue samples of BC obtained from male patients had a higher α diversity, although the urine samples did not show any difference [76]. 

Similarly, another study found that no differences in α or β diversity could be confirmed between the urine samples obtained from both male and female patients [81]. Nevertheless, the same study found that female patients tended to show a higher differential abundance of Bacteriodetes (genus *Lactobacillus*, *Actinotignum*, *Prevotella*, *Vellionella*, *Campylobacter*, and *Enterococcus*), while male patients had a higher relative abundance of Actinobacteria (genus *Pelomonas*, *Corynebacterium*, *Finegoldia*) [81]. Abundance of *Veillonella* and *Corynebacterium* was also confirmed [86], alongside abundance of *Acinetobacter*, *Anaerococcus*, and *Rubrobacter* [85].

Although sex differences in the urinary microbiome have been observed in some of the studies performed thus far, a more extensive investigation is required to confirm the influence of sex hormones on the diversity of urinary microbiota and to confirm whether those differences are responsible for the different disease progression in males and females.

**Table 3 ijms-25-01732-t003:** Urinary microbiome in bladder cancer.

Sex (M/F)	BC/HC	Sample	Increase	Decrease	Reference
NA	8/6	urine	*Streptoccocus*, *Pseudomonas*, *Anaerococcus*	NA	[78]
49M	31/18	urine	*Acinetobacter*, *Anaerococcus*, *Sphingobacterium*, *Geobacillus*	*Serratia*, *Proteus*, *Roseomonas*, *Ruminiclostridium-6*, *Eubacterium-x*	[85]
33M	12/11	urine	*Fusobacterium nucleatum*, *Actinobaculum*, *Facklamia*, *Campylobacter*, *Subdoligranulum*, Ruminococcaceae	*Veillonella*, *Streptococcus*, *Corynebacterium*	[80]
35M/20F	29/16	urine	*Actinomyces*	*Streptococcus*, *Bifidobacterium*, *Lactobacillus*, *Veillonella*	[103]
18M/6F	24/0	urine	Enterobacteriaceae, *Streptococcus*, *Lactobacillus*, *Ureaplasma*, *Corynebacterium*, *Stenotrophomonas*, *Enterococcus*, *Staphylococcus*	X	[104]
41M	8/33	urine	No differences	No differences	[105]
42M/6F	38/10	urine	*Bacteroides*, *Faecalibacterium*	*Lachnoclostridium*, Burkholderiaceae	[83]
14M/8F	22/0	urine	*Tepidimonas (male)*	*Prevotella*, *Veillonella* (male)	[106]
NA	62/19	urine	*Micrococcus*, *Brachybacterium*	X	[107]
NA	15/11	urine	Bacteroidaceae, E*rysipelotrichales*, Lachnospiraceae	X	[108]
36M/7F	43/10	urine	MIBC: *Haemophilus*, *Veillonela* NMIBC: *Cupriavidus*, *Serratia*, *Brochothrix*, *Negativicoccus*, *Escherichia-Shigella*, *Pseudomonas*	X	[81]
40M	40/0	urine	*Pseudomonas*, *Staphylococcus*, *Corynebacterium*, *Acinetobacter*	X	[109]
56M	32/24	urine	*Arthrobacter ginkgonis*, *Micrococcus* sp., *Hydrogenophaga aquatica*, *Defluviimonas pyrenivorans*, *Propionibacterium * *namnetense*, *Corynebacterium halotolerans*, *Acinetoacter celticus*	X	[110]
61M	51/10	urine	*Veillonella*, *Corynebacterium*	*Ruminococcus*	[86]
5M/5F	10/0	tissue and urine	Tissue: *Akkermansia*, *Bacteroides*, *Clostridium sensu stricto*, *Enterobacter*, *Klebsiella*	X	[76]
70M/38F	49/59	tissue and urine	Tissue: *Burkholderia*; Urine (Male): *Acidobacteria*, Opitutaceae; Urine (Female): *Klebsiella*	Urine (Male): Tissierellaceae, *Alphaproteobacteria*, *Rhizobiales*, *Sphingomonadales*, *Pasteurellales*, Streptococcaceae, Corynebacteriaceae, Patulibaceteraceae; Urine (Female): *Betaproteobacteria*, *Burkholderiales*, *Pseudomonadales*, Comamonadaceae, Moraxellaceae, Coriobacteriaceae, *Coriobacteriia*	[75]
22M	22 BC tissues 12 healthy tissue	tissue	*Cupriavidus* spp., *Acinetobacter*, *Anoxybacillus*, *Escherichia*, *Shigella*, *Geobacillus*, *Pelomonas*, *Ralstonia*, *Sphingomonas*	*Lactobacillus*, *Prevotella*, *Ruminococcus*	[82]

NA = not available; X = no decrease in the density of specific bacteria was recorded.

## 4. Microbiome, Inflammation, and Bladder Cancer 

It is well-known that inflammation or local or systemic proinflammatory conditions can play a pivotal role in cancer development, metastasis, and drug resistance. Specifically, infections activate acute inflammations that can turn into chronic inflammation due to the persistent activation of host defense mechanisms against microbial infection or cellular injury, thus paving the way for carcinogenesis [111].

There is a link between imbalances (or deviations) in the microbiome, also referred to as dysbiosis, and a higher abundance of opportunistic pathogens, which, in turn, increase the antigen load sustaining chronic inflammation [112]. Urinary tract infections have been pointed out as a significantly elevated risk factor of BC [113]. Accordingly, *Schistosoma* infection serves as an oncogenic factor in BC by stimulating the synthesis of N-nitrosamine and sustaining the induction of chronic inflammation [114,115].

In BC, the recruitment of inflammatory cells, including myeloid-derived suppressor cells (MDSCs), regulatory T cells, dendritic cells, mast cells, neutrophils, and lymphocytes, is associated with the release of multiple proinflammatory molecules, matrix metalloproteinases (MMPs) cytokines/chemokines, and the activation of signaling pathways in TME that may elicit BC tumorigenesis [116].

The persistence of chronic inflammation damages epithelial cells through multiple mechanisms: the generation of reactive oxygen (ROS) and nitrogen species (RNS), consequently leading to cell death and further destruction of the epithelial barrier [117,118].

A systemic inflammatory marker, the serum neutrophil:lymphocyte ratio (NLR) can be predictive of disease recurrence and progression in BC, with the correlation being particularly strong in MIBC [119,120,121,122].

Evidence suggests that dysbiosis mediated by urinary tract infections can lead to enhanced TNF-α secretion, which may play a critical role in the development of BC [123,124,125]. Higher TNF-α serum levels in BC patients with or without schistosomiasis have been reported [126]. Likewise, increased TNF-α expression has been detected in tumor tissue compared to healthy urothelium, as well as in T3 and T4 advanced-stage patients compared to early-stage patients [126,127]. Commensal probiotic bacteria may exert an anti-inflammatory effect by reducing the TNF-α levels [123,124].

IL-6 is another critical participant in the inflammatory process associated with a more advanced clinical stage and a lower survival rate of BC [128,129,130]. High IL-6 levels sustain the immunosuppressive TME by increasing the proliferation of MDSCs in MIBC tissue, which, in turn, inhibits the proliferative capacity of CD4^+^ or CD8^+^ T cells [131]. Moreover, the Signal Transducer and Activator of Transcription 3 (STAT3) targeted by IL-6 has been found to be highly activated in infiltrating basal-type urothelial bladder cancer tissue compared to luminal UBC [132]. Accordingly, by reducing IL-6 levels or by employing anti-IL-6 mAb therapy, it may be possible to decrease the inflammatory process and increase the effectiveness of the treatment of BC [133,134,135].

An association between high IL-23R and IL-17 expression levels in the tumor tissue and serum of patients with BC and a poor prognosis has been established [136,137]. IL-17 activates Stat3 in tumor and tumor stromal cells through an IL-6-dependent mechanism, upregulating prosurvival and proangiogenic genes [138]. Targeting the IL-23/IL-17 inflammatory pathway may reduce bladder urothelial carcinoma risk by reducing the T-helper 17 cell-mediated inflammatory process [139]. Evidence suggests that the enterotoxigenic *Bacteroides fragilis*, an enterotoxin-producing bacterium in the gastrointestinal tract, may be more prevalent in colorectal cancer patients compared to healthy individuals and may induce colon carcinogenesis by activating IL-17 [140]. Gut microbiota-derived metabolites, particularly propionate, have been shown to reduce IL-17 and IL-22 production by intestinal γδT cells in a histone deacetylase-dependent manner [141]. Studies on the gut microbiota and colorectal cancer suggest the importance of elucidating the role that the urinary microbiome plays in the IL-17 pathway and in the onset and progression of BC.

Depending on their properties and polarization, tumor-associated macrophages (TAMs) can be associated with either poor or improved prognosis in relation to BC. TAMs, displaying the M2 phenotype, promote BC metastasis by producing proangiogenic and immunosuppressive cytokines [142,143,144]. A significant increase in TAM counts has been associated with a worse outcome and identified as a useful indicator for predicting the response of BC to in situ intravesical BCG instillation before treatment initiation [145,146]. Accordingly, the predominance of immunosuppressive M2 macrophages in the stroma of low-hypoxia BC has been shown to have a connection with BCG treatment outcome [142]. Targeting the major inflammatory regulator, namely the necrosis factor kappa B (NF-κB), has been found to suppress both oncogenic and metastatic potential in BC cells while preventing the M2 polarization of TAMs [147].

A key component with a major role in establishing an immunosuppressive milieu in BC is the higher numbers of MDSCs in BC tissue compared to controls [148,149,150]. Zhang et al. showed that MDSCs from fresh BC tissues displayed high levels of suppressive molecules, such as Arg1, iNOS, ROS, PD-L1, and P-STAT3, and stronger suppression of T-cell proliferation, and further identified the CXCL2/MIF-CXCR2 axis as an important mediator in MDSC recruitment and as a potential therapeutic target in BC patients [151]. Bladder cancer cells have further been shown to activate the COX2/mPGES1/PGE_2_ pathway involved in the upregulation of MDSCs and the expression of PD-L1 [152]. Hence, it is imperative to define the role that urine microbiota plays in bladder inflammation, as this could potentially generate innovative approaches for the prevention and management of bladder cancer.

## 5. Microbiome and Bladder Cancer Therapy

Emerging evidence suggests that the microbiota and its bioactive metabolites may regulate the immunological microenvironment and influence the therapy for bladder cancer.

### 5.1. Microbiome and BCG Responsiveness

The bacterial vaccination strain (BCG) is the gold standard adjuvant therapy for high-risk NMIBC, and it also represents an option for the treatment of intermediate-risk NMIBC [153,154]. Despite being used successfully for more than 40 years, a range of adverse side effects have been reported [155]. Deciphering the precise mechanisms underlying the interaction between the microbiome and other host-related factors may increase its efficacy and minimize adverse effects.

Although it is still unclear how BCG prevents tumor recurrence, it has been suggested that BCG stimulates a proinflammatory innate immune response, which, in turn, triggers an adaptive immune response against both BCG and tumor antigens [156,157,158]. BCG produces several pathogen-associated molecular patterns (PAMPs), recognized by host pattern recognition receptors (PRRs), that activate the production of proinflammatory cytokines [159,160,161]. Effective BCG therapy remodels the bladder TME to display proinflammatory features through the release of proinflammatory cytokines, and the degree of immune response, assessed by urinary cytokine quantification, has been associated with rates of recurrence and progression [162,163]. A direct interaction between BCG and bladder and urothelial cells triggers the innate immune response by activating polymorphonuclear cells (PMNs) [164], dendritic cells [165], and NK cells [166]. BCG triggers the activation of both CD4+ and CD8+ T cells into the bladder wall, eliminating tumor cells and, consequently, decreasing tumor recurrence [167]. Several studies have reported an association between advanced age and poorer outcomes in patients treated with BCG for NMIBC [168,169,170]. Since immunosenescence attenuates the innate and adaptive immune systems in elderly patients, age has been identified as an independent risk factor in relation to the efficacy of BCG therapy [168,169,170]. However, there is a high heterogeneity among studies due to small non-randomized studies or adherence of elderly patients to intravesical BCG treatment because of toxicity concerns. Notably, recently, it has been found that, in a cohort of patients with NMIBC treated with adequate BCG, age >70 years was not a predictor of poor outcomes or associated with adverse oncological events. BCG should not be withheld from older patients seeking bladder-sparing options [171]. In addition to triggering innate and adaptive immunity, BCG exerts direct cytotoxic effects on BC cells, activating apoptosis through the caspase 8, caspase 9, and proapoptotic protein BID signaling pathway [172,173]. 

However, about 30% of patients experience tumor recurrence within the first three years following BCG therapy [157]. In a study by Kates et al. comparing BCG responders with non-responders, it was shown that tumor PD-L1 expression predicts an unfavorable BCG response [158].

Early data suggest that the commensal urinary microbiome may have an impact on efficacy and may predict the response to BCG immunotherapy in NMIBC patients [168,170,174]. A difference in urine microbiota has been reported between BCG responders and non-responders at the OTU level in one investigation involving 31 patients with high-risk NMIBC receiving BCG therapy. The phylum Firmicutes, including *Lactobacillus* and urinary *Proteobacteria*, was found to be prevalent in the urine of disease-free patients compared to the urine of patients who experienced a recurrence of the disease [174]. In preliminary work by Knorr et al., a differential enrichment in *Corynebacterium* and *Pseudomonas* was found in BCG responders, suggesting that this therapy may alter the composition of the local microbiome and that *Corynebacterium* may act as a pretreatment marker for favorable BCG response [175]. In contrast to their previous findings, another recent analysis using shotgun metagenomics and 16S next-generation sequencing (NGS) has reported an increase in *Corynebacterium* in BCG non-responders [176]. Notably, the authors demonstrated enrichment of *Lactobacillus* spp. in BCG responders, underlining consistent perturbations in urinary tract microbiota between BCG responders and non-responders [176].

An investigation involving patients with NMIBC has found that non-recurrence patients tend to exhibit higher levels of *Escherichia/Shigella* and *Ureaplasma*, while patients experiencing a relapse after BCG or intravesical therapy tend to exhibit higher levels of *Aerococcus*, thus being predictive of a poor cancer response [177].

The interplay between the urine microbiome and BCG response is complex: it may either reduce the effectiveness of BCG by inhibiting the cytotoxic response or it might have a synergistic impact (i.e., Lactobacilli) by inducing cytotoxic and antiproliferative effects which would increase BCG anticancer activity [178].

To overcome limitations associated with attempts to identify a link between BCG response and the urine microbiota, controlled large-scale multicenter studies with well-defined methodologies for the collection of samples from either intravesical or post-urethral urine are needed. In this respect, a prospective observational study of the role played by the microbiome in BCG responsiveness prediction (NCT05204199) has been approved [179]. In addition, another ongoing clinical trial (SILENTEMPIRE, NCT05204199) will investigate the microbial profiles from the bladder and the feces of NMIBC patients as a predicting tool for therapy response prior to BCG administration. The study will attempt to establish a correlation between the BCG response and the microbiome profile.

### 5.2. Microbiome and Immune Checkpoint Immunotherapy

Immune checkpoint inhibitors targeting the programmed death 1/programmed death-ligand 1 (PD-1/PD-L1) and cytotoxic T-lymphocyte-associated protein 4 (CTLA-4) pathways have shown significant antitumor activity, tolerable safety profiles, and durable, long-term responses in relation to different cancers, including in metastatic urothelial carcinoma patients [180,181,182].

Evidence suggests that resistance to ICIs can be attributed to abnormal gut microbiome composition, as PD-L1 expression is modulated by microbiota [183,184,185,186,187]. As such, specific bladder microbial compositions may be linked to the response of bladder cancer to anti-PD-1/PD-L1 therapy. 

Little is understood about the bladder microbiome itself and about whether it alters the tumor microenvironment and, as a consequence, the response to PD1 inhibitors. Conversely, a significant association between a higher abundance of *Akkermansia*, *Enterococcus*, *Clostridiales*, *Ruminococcaceae*, *Faecalibacterium*, and *Bifidobacterium* and an improved response to PDL1 therapy in cancers such as lung, renal, and melanoma has been found [183,185,186,187,188].

NMIBC patients expressing high PD-L1 levels have a poor prognosis and relapse-free survival.

A study by Chen at al. suggested a relationship between urogenital microbiota and PD-L1 expression in male NMIBC patients. Different compositions of urogenital microbiota were found between the PD-L1-positive and the PD-L1-negative groups. An increase in the abundance of some bacterial genera (e.g., *Leptotrichia*, *Roseomonas*, and *Propionibacterium*) and a decrease in the abundance of others (e.g., *Prevotella* and *Massilia*) were observed in the PD-L1-positive group compared to the negative one [189]. 

The use of antibiotics has been associated with poorer outcomes in cancer patients treated by ICIs, reporting a decrease in overall survival, progression-free survival, and disease control [190,191,192,193]. Treatment with antibiotics may modify the gut microbiota, thus modifying bacteria composition or their metabolites that affect antitumor immunity, inflammation, and ICI responses [194]. According to a recent study by Raggi et al., antibiotic use is associated with a lower benefit from neoadjuvant immunotherapy for bladder cancer [195]. Successful neoadjuvant immunotherapy may be correlated with the diversity of the urine microbiota. 

Improved understanding of gut and urinary tract microbiota and its effects on priming immune cells against specific antigens might shed some light on potential targets for therapeutic intervention and might improve the response to these cutting-edge immunotherapies.

## 6. Conclusions

Despite the advancements in cancer understanding and diagnostic methods, BC continues to pose a formidable challenge characterized by high recurrence rates and poor prognosis, especially in advanced stages. Recent research has shed light on the potential influence of the genitourinary microbiota on therapeutic responses in BC systemic treatment.

The characteristics of the healthy urinary microbiome, although less abundant and diverse than at other sites, exhibit significant interindividual variability, showcasing microbial diversity in both males and females. In healthy women, various urotypes have been observed, with the prevalence of *Lactobacillus* and *Gardnerella*. Additional urotypes include *Prevotella*, *Corynebacterium*, *Sneathia*, *Streptococcus*, and *Escherichia*, with age-related differences in bacterial phyla dominance. Early investigations into the healthy male urinary microbiome have identified genera such as *Lactobacillus*, *Sneathia*, *Veillonella*, *Corynebacterium*, *Prevotella*, *Streptococcus*, and *Ureaplasma*. 

Dysbiosis in the urinary microbiota has been associated with urinary diseases, including UTIs, interstitial cystitis, urinary incontinence, and bladder pain syndrome, most frequently associating bacteria, such as *Escherichia coli*, *Enterococcus*, *and Staphylococcus*, with the abovementioned pathologies. Nevertheless, the exact causative agents remain unknown. Despite numerous studies attempting to link the prevalence of certain bacteria to BC, challenges persist in drawing definitive conclusions. The lack of coherence among study results, as seen in varied α and β diversity findings, complicates our understanding. The multifaceted nature of microbiome research, coupled with the complexity of BC, requires further investigation and larger-scale studies to unravel the intricate associations. 

As of today, several bacteria have been proposed as potential biomarkers for BC, including *Cupriavidus* spp., Brucellaceae, *Acinetobacter*, *Anoxybacillus*, *Escherichia-Shigella*, *Geobacillus*, *Pelomonas*, *Ralstonia*, and *Sphingomonas*, but larger studies are needed for validation. Chronic inflammation, fostered by persistent immune system activation against microbial threats or cellular damage, supports carcinogenesis, with microbiome dysbiosis contributing to this chronic inflammatory environment conducive to BC.

Therapeutic approaches for BC involve BCG as the gold standard adjuvant therapy for NMIBC, stimulating a proinflammatory immune response. However, challenges arise due to several adverse effects, with the commensal urinary microbiome possibly impacting BCG efficacy. Immune checkpoint immunotherapy targeting PD-1/PD-L1 and CTLA-4 pathways demonstrates significant antitumor activity, with resistance linked to abnormal gut microbiome composition.

Additional research is essential to delve deeper into the correlation among bladder cancer, microbiome dysbiosis, and chronic inflammation. These investigations could potentially open avenues for the development of innovative strategies for BC prevention, novel treatment approaches, and enhanced tools for risk stratification.

## Figures and Tables

**Figure 1 ijms-25-01732-f001:**
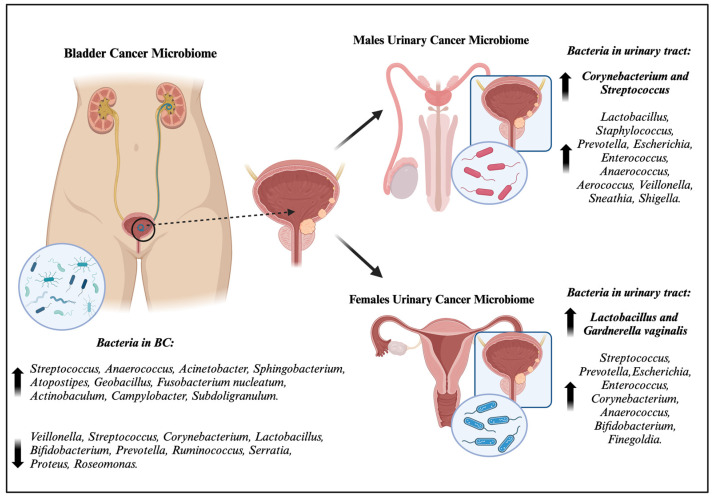
Characterization of bladder cancer microbiome and differences between male urinary microbiome and female urinary microbiome. Up arrows = increase in abundance; down arrows = decrease in abundance. Figure created with http://biorender.com (accessed on 1 January 2024).

**Table 1 ijms-25-01732-t001:** Microbiome composition of urine in healthy females.

Age	No	Bacteria	Urine Sample	Reference
27–67	8	*Lactobacillus*, *Prevotella*, *Gardnerella*, *Peptoniphilus*, *Dialister*, *Finegoldia*, *Anaerococcus*, *Allisonella*, *Streptococcus*, *Staphylococcus*	MSU	[31]
22–51	15	*Streptococcus*, *Aerococcus*, *Gardnerella*, *Prevotella*, *Escherichia*, *Enterococcus*	MSU	[38]
NA	12	*Lactobacillus*, *Actinobaculum*, *Aerococcus*, *Anaerococcus*, *Atopobium*, *Burkholderia*, *Corynebacterium*, *Gardnerella*, *Prevotella*, *Ralstonia*, *Sneathia*, *Staphylococcus*, *Streptococcus*, *Veillonella*	MSU SPA TUC	[39]
NA	10	*Firmicutes*, *Actinobacteria*, *Bacteroidetes*	MSU	[34]
NA	24	*Lactobacillus*, *Corynebacterium*, *Streptococcus*, *Actinomyces*, *Staphylococcus*, *Aerococcus*, *Gardnerella*, *Bifidobacterium*, *Actinobaculum*	TUC	[30]
35–65	58	*Lactobacillus*, *Gardnerella*, *Corynebacterium*, Enterobacteriaceae, *Anaerococcus*, *Bifidobacterium*, *Streptococcus*, *Staphylococcus*, *Sneathia*, *Peptoniphilus*, *Atopobium*, *Rhodanobacter*, *Trueperella*, *Alloscardovia*, *Veillonella*	TUC	[27]
18–25	24	*Lactobacillus*, *Staphylococci*, *Peptococcus*, *Corynebacterium*, *Propionibacterium*, *Eubacterium*, *Peptostreptococcus*, *Candida*, *Bacteroides*, *Bacillus*, *Veillonella*, Enterobacteriaceae, *Staphylococcus aureus*, *Enterococcus*, *Micrococcus*, *Prevotella*, *Actinomyces*, *Streptococcus*	MSU	[40]
35–65	60	*Lactobacillus*, *Gardnerella*, *Staphylococcus*, *Streptococcus*, *Enterococcus*, *Bifidobacterium*, *Atopobium*, Enterobacteriaceae	TUC	[41]
NA	10	*Anoxybacillus*, *Lactobacillus*, *Prevotella*, *Gardnerella*, *Arthrobacter*, *Escherichia*, *Shigella*	TUC	[42]
19–62	49	*Prevotella amnii*, *Gardnerella vaginalis*, *Atopobium vaginae*, *Lactobacillus iners*, *Shigella sonnei*, *Escherichia coli*, *Enterococcus faecalis*, *Streptococcus agalacticie*, *Citrobacter murliniae*, *Lactobacillus crispatus*	MSU	[36]
NA	10	*Lactobacillus*, *Corynebacterium*, *Gardnerella*, *Prevotella*, *Bacillus*	MSU	[32]
NA	60	*Staphylococcus epidermidis*, *Micrococcus luteus*, *Lactobacillus gasseri*, *Escherichia coli*, *Streptococcus oralis*, *Neisseria perflava*, *Lactobacillus crispatus*, *Lactobacillus jensenii*, *Gardnerella vaginalis*, *Rothia mucilaginosa*, *Lactobacillus delbrueckii*, *Lactobacillus rhamnosus*, *Bacillus infantis*, *Actinomyces odontolyticus*, *Bacillus idriensis*, *Corynebacterium amycolatum*, *Streptococcus anginosus*, *Streptococcus agalactiae*, *Gordonia terrae*, *Staphylococcus warneri*, *Lactobacillus iners*, *Streptococcus mitis*, *Bifidobacterium bifidum*, *Streptococcus gordonii*, *Aerococcus urinae*, *Actinomyces neuii*, *Enterococcus faecalis*, *Streptococcus salivarius*, *Streptococcus sanguinis*, *Corynebacterium aurimucosum*, *Actinomyces naeslundii*, *Streptococcus equinus*, *Alloscardovia omnicolens*, *Corynebacterium tuscaniense*, *Bifidobacterium longum*	TUC	[29]
≈53	84	*Lactobacillus*, *Streptococcus*, *Tepidimonas*, *Prevotella*, *Flavobacterium*, *Escherichia*, *Ureaplasma*, *Shuttleworthia*, *Aerococcus*, *Gardnerella*, *Veillonella*, *Bacteroides*, *Enterobacter*, *Acidovorax*, *Sneathia*, *Clostridium*, *Fusobacterium*, *Sphingobium*, *Proteus*, *Trabulsiella*	TUC	[43]
NA	224	Predominant urotypes: *Lactobacillus*, *Gardnerella*, *Streptococcus*, *Escherichia*Other: *Aerococcus*, *Alloscardovia*, *Anaerococcus*, *Bifidobacterium*, *Corynebacterium*, *Enterococcus*, *Finegoldia*, *Klebsiella*, *Prevotella*, *Staphylococcus*	TUC	[44]
≈38	110	*Bifidobacterium*, *Staphylococcus*, *Lactobacillus*, *Corynebacterium*	MSU	[45]

NA = not available; MSU = midstream urine; SPA = suprapubic aspirate; TUC = transurethral catheter.

**Table 2 ijms-25-01732-t002:** Microbiome composition of urine from healthy males.

Age	No	Bacteria	Urine Sample	Reference
≈18	9	*Lactobacillus*, *Corynebacterium*, *Escherichia*, *Streptococcus*	FC	[37]
≈28	22	*Lactobacillus*, *Sneathia*, *Veillonella*, *Corynebacterium*, *Prevotella*, *Streptococcus*, *Ureaplasma*, *Mycoplasma*, *Anaerococcus*, *Atopobium*, *Aerococcus*, *Staphylococcus*, *Gemella*, *Enterococcus*, *Finegoldia*	FC	[54]
24–50	11	*Lactobacillus*, *Klebsiella*, *Corynebacterium*, *Staphylococcus*, *Streptococcus*, *Aerococcus*, *Gardnerella*, *Prevotella*, *Escherichia*, *Enterococcus*	MSU	[38]
14–17	18	*Corynebacterium*, *Lactobacillus*, *Staphylococcus*, *Gardnerella*, *Streptococcus*, *Anaerococcus*, *Veillonella*, *Prevotella*, *Escherichia*	FC	[55]
39–86	6	*Firmicutes*	MSU	[34]
18–25	28	Staphylococci, *Eubacterium*, *Corynebacterium*, *Peptostreptococcus*, *Enterococcus*, *Bacteroides*, *Peptococcus*, *Megasphaera*, *Mobiluncus*, Enterobacteriaceae, *S. aureus*, *Propionibacterium*, *Veillonella*, *Fusobacterium*	MSU	[40]
23–58	31	*Prevotella amnii*, *Sneathia amnii*, *Shigella sonnei*, *Enterococcus faecalis*, *Streptococcus agalacticie*, *Citrobacter murliniae*	MSU	[36]
NA	10	*Streptococcus*, *Lactobacillus*, *Prevotella*, *Corynebacterium*, *Pseudomonas*	MSU	[32]
≈43	97	*Staphylococcus*, *Propionibacterium*	MSU	[45]

NA = not available; FC = first catch; MSU = midstream urine.

## Data Availability

Not applicable.

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
