# Peer review of "The Urinary Microbiome in Health and Disease: Relevance for Bladder Cancer"

_ijms, 2024, doi:10.3390/ijms25031732_

Round 1

Reviewer 1 Report

Comments and Suggestions for Authors

The manuscript provides a comprehensive overview of the role of urinary microbiota in BC. m highlighting advancements in DNA sequencing and microbiota testing. The emphasis on potential biomarkers for risk stratification and therapeutic response is commendable.

The authors should reconsider the introduction regarding bladder tumors, as some information appears questionable. For instance chemotherapy is typically administered in a neoadjuvant setting requires clarification. Additionally, the claim regarding the low success rate of Immune Checkpoint Inhibitors is questionable.

Provide a Reference for the statement in lines 72-73

It is crucial to underscore the role of the inflammatory response BCG  efficacy. Notably, certain studies have indicated that age may not be a significant risk factor for BCG efficacy line 474  (see https://doi.org/10.1111/bju.16127 ).

Author Response

We are pleased to submit the revised version of our manuscript entitled “The urinary microbiome in health and disease: relevance for bladder cancer, by Kustrimovic et al, to be considered for the publication in International Journal of Molecular Sciences.

We have addressed the reviewer’s comments. You can find a point-by-point response to reviewer comments.

Reviewer #1: The manuscript provides a comprehensive overview of the role of urinary microbiota in BC highlighting advancements in DNA sequencing and microbiota testing. The emphasis on potential biomarkers for risk stratification and therapeutic response is commendable.

  • The authors should reconsider the introduction regarding bladder tumors, as some information appears questionable. For instance, chemotherapy is typically administered in a neoadjuvant setting and requires clarification. Additionally, the claim regarding the low success rate of Immune Checkpoint Inhibitors is questionable.

We thank the reviewer for their insightful comments and suggestions aimed at enhancing the quality of our review. In response, we have followed these recommendations and enriched the introduction with additional information regarding bladder tumors.

  • Provide a Reference for the statement in lines 72-73

We have provided a reference for the statement in lines 72-73.

  • It is crucial to underscore the role of the inflammatory response BCG efficacy. Notably, certain studies have indicated that age may not be a significant risk factor for BCG efficacy line 474 (see https://doi.org/10.1111/bju.16127).

We thank the reviewer for their suggestion and for highlighting the role of the inflammatory response in BCG efficacy. We have included more articles in the manuscript to discuss the role of the inflammatory response and its association with age in relation to BCG efficacy.

Reviewer 2 Report

Comments and Suggestions for Authors

In the article titled "The urinary microbiome in health and disease: relevance for bladder cancer ", the authors present information about the urobiome and its potential role in the development of bladder cancer. The paper contains an overview of information regarding the analysis of the microbiome composition in healthy people and cancer patients. It is worth emphasizing here that the authors collected so much information about the urobiome of male, which was a challenge because this microbiome has not yet been intensively studied. In my opinion, the most interesting element of the article is the one concerning determining the influence of the microbiome on the urinary tract environment, which could influence the development of cancer. The article is well thought out and planned. The figures are prepared flawlessly and are referenced in the text. Due to my duty as a reviewer, I will only mention some minor errors in the text. For example: family names, e.g. Enterobacteriaceae, should not be written in italics (Table 1,2,3); Klebsiella pneumonia (line 341) it shoul be pnemoniae; urinary (line 479) should not be written in italic; spp. not in italic; Lactobacilli (line 496) should be lactobacilli without italic.

Author Response

We are pleased to submit the revised version of our manuscript entitled “The urinary microbiome in health and disease: relevance for bladder cancer, by Kustrimovic et al, to be considered for the publication in International Journal of Molecular Sciences.

We have addressed the reviewer’s comments. You can find a point-by-point response to reviewer comments.

Reviewer #2:

In the article titled "The urinary microbiome in health and disease: relevance for bladder cancer ", the authors present information about the urobiome and its potential role in the development of bladder cancer. The paper contains an overview of information regarding the analysis of the microbiome composition in healthy people and cancer patients. It is worth emphasizing here that the authors collected so much information about the urobiome of male, which was a challenge because this microbiome has not yet been intensively studied. In my opinion, the most interesting element of the article is the one concerning determining the influence of the microbiome on the urinary tract environment, which could influence the development of cancer. The article is well thought out and planned. The figures are prepared flawlessly and are referenced in the text. Due to my duty as a reviewer,

I will only mention some minor errors in the text. For example: family names, e.g. Enterobacteriaceae, should not be written in italics (Table 1,2,3); Klebsiella pneumonia (line 341) it should be pnemoniae; urinary (line 479) should not be written in italic; spp. not in italic; Lactobacilli (line 496) should be lactobacilli without italics.

We sincerely appreciate the reviewer's positive comments and feedback. We have introduced the requested modifications.

Round 2

Reviewer 1 Report

Comments and Suggestions for Authors

the authors responded satisfactorily to the reviewer's comments